# Remodeling of the *Histoplasma Capsulatum* Membrane Induced by Monoclonal Antibodies

**DOI:** 10.3390/vaccines8020269

**Published:** 2020-06-02

**Authors:** Meagan C. Burnet, Daniel Zamith-Miranda, Heino M. Heyman, Karl K. Weitz, Erin L. Bredeweg, Joshua D. Nosanchuk, Ernesto S. Nakayasu

**Affiliations:** 1Biological Sciences Division, Pacific Northwest National Laboratory, Richland, WA 99352, USA; Meagan.Burnet@pnnl.gov (M.C.B.); hmheyman@gmail.com (H.M.H.); Karl.Weitz@pnnl.gov (K.K.W.); 2Department of Microbiology and Immunology and Division of Infectious Diseases, Department of Medicine, Albert Einstein College of Medicine, Bronx, NY 10461, USA; daniel.zamithmiranda@einstein.yu.edu; 3Environmental and Molecular Sciences Division, Pacific Northwest National Laboratory, Richland, WA 99352, USA; erin.bredeweg@pnnl.gov

**Keywords:** *Histoplasma capsulatum*, antibody, lipids, metabolism, multi-omics

## Abstract

Antibodies play a central role in host immunity by directly inactivating or recognizing an invading pathogen to enhance different immune responses to combat the invader. However, the cellular responses of pathogens to the presence of antibodies are not well-characterized. Here, we used different mass spectrometry techniques to study the cellular responses of the pathogenic fungus *Histoplasma capsulatum* to monoclonal antibodies (mAb) against HSP60, the surface protein involved in infection. A proteomic analysis of *H. capsulatum* yeast cells revealed that mAb binding regulates a variety of metabolic and signaling pathways, including fatty acid metabolism, sterol metabolism, MAPK signaling and ubiquitin-mediated proteolysis. The regulation of the fatty acid metabolism was accompanied by increases in the level of polyunsaturated fatty acids, which further augmented the degree of unsaturated lipids in *H. capsulatum*’s membranes and energy storage lipids, such as triacylglycerols, phosphatidylcholines, phosphatidylethanolamines and phosphatidylinositols. MAb treatment also regulated sterol metabolism by increasing the levels of cholesterol and ergosterol in the cells. We also showed that global changes in the lipid profiles resulted in an increased susceptibility of *H. capsulatum* to the ergosterol-targeting drug amphotericin B. Overall, our data showed that mAb induction of global changes in the composition of *H. capsulatum* membranes can potentially impact antifungal treatment during histoplasmosis.

## 1. Introduction

Antibodies are major components of the humoral immune response and primarily fight infections by facilitating phagocytosis (opsonization), activating the complement system, neutralizing or lysing pathogens or mediating specific cytotoxic responses [1]. In fungal infections, antibodies against components of fungal cells, such as capsular polysaccharide, β-glucans, glycolipids, melanin and proteins, can have protective roles during infections. Antibodies against glucoroxylomann (GXM), the main component of *Cryptococcus neoformans* capsule, inhibit the release of polysaccharide from the fungus [2] and also hydrolyze glycans and peptides [3]. Other antibodies, such as an anti-monohexosylceramide antibody, have growth-inhibitory activity on *C. neoformans* [4]. Recently, engineered chimeric antibodies have been shown to confer pan-fungal protection by targeting chitin, a conserved polysaccharide present across the Fungi kingdom [5]. In *Histoplasma capsulatum*, antibodies against the cell-wall proteins, histone 2B-like protein and 60 kDa heat shock protein (HSP60), protect by opsonizing and altering the intracellular fate of the fungus inside macrophages, leading to the production of Th1-type cytokines [6,7]. 

Although antibodies have many known roles in terms of protecting the host against infection, fungal responses to antibodies are largely unknown. McClelland et al. showed that antibodies against *C. neoformans*’ capsule regulate gene expression in yeast, modulating its metabolism. They also observed an increase in the synthesis of fatty acids and genes related to the ergosterol biosynthetic pathway. Furthermore, antibody treatment increased fungal susceptibility to amphotericin B, which targets ergosterol on the fungal membrane [8]. We selected two monoclonal antibodies (mAb) that bind the same HSP60 epitope on the surface of *H. capsulatum*. Treatment of mice with these mAbs leads to different outcomes in infection with *H. capsulatum*. 6B7 (IgG1) confers significant protection in a lethal histoplasmosis model in mice, whereas 7B6 (IgG2b) is not protective [7]. More recently, Baltazar et al. showed that these mAbs induced changes in the size and composition of extracellular vesicles secreted by *H. capsulatum*. These changes were not restricted to the total protein amount in the vesicles, as mAbs also affected the levels of virulence factors, such as phosphatase, laccase and catalase [9,10]. Moreover, although the two mAbs tested recognized the exact same epitope in HSP60, they differentially modulated the size and composition of the vesicles [9,10], consistent with one being protective and the other non-protective.

In the present paper, we studied the cellular effects of HSP60-binding mAbs 6B7 and 7B6 to *H. capsulatum* yeast cells by a multi-omics approach. Our results showed that antibodies induced global changes in the lipid composition of *H. capsulatum’s* membranes. Such changes increased the susceptibility of *H. capsulatum* cells to the ergosterol-targeting drug amphotericin B, which suggests that the antibodies may impact the efficacy of antifungal treatment during histoplasmosis.

## 2. Materials and Methods 

### 2.1. Cell Culture and Treatment 

*H. capsulatum* G217B strain was purchased from the American Type and Culture Collection (ATCC, Manassas, VA, USA). Yeast cells were cultivated in 4 biological replicates in Ham’s F12 medium at 37 °C as previously described [10]. An additional well-characterized mAb, 18B7, that binds to *C. neoformans* GXM [11] was used as a possible control. MAb (6B7, 7B6 and 18B7) produced by hybridoma cells were quantified by ELISA [7,11]. Ten milliliters of *H. capsulatum* culture were incubated in the presence or absence of each mAb at a final concentration of 10 μg/mL for 24 h at 37 °C. Cells were then washed with phosphate-buffered saline (PBS) and frozen until the respective extractions. These samples were prepared and analyzed in parallel to the dataset described by Zamith-Miranda et al. [12].

### 2.2. Sample Extraction

#### 2.2.1. Metabolite, Protein, and Lipid Extraction (MPLEx) 

Samples were processed using the metabolite, protein and lipid extraction (MPLEx) method as described [13]. Briefly, metabolites, proteins and lipids were simultaneously extracted by suspending each *H. capsulatum* cell pellet with 200 μL of 50 mM NH_4_HCO_3_ followed by transfer to a new microcentrifuge tube containing 100 μL of 0.1 mm zirconia/silica beads. The cell pellet was vortexed for 10 × 30 s with 1 min intervals on ice, and the supernatant was transferred to new 2 mL microcentrifuge tubes (Sorenson). The beads were washed with 200 μL of 50 mM NH_4_HCO_3_ and the lysates were combined before extraction. To the 360 μL of recovered lysate, 1200 μL of −20 °C chloroform:methanol (C:M) (2:1) was added and incubated for 5 min on ice. The sample was then vortexed for 1 min before being subjected to centrifugation of 12,000 rpm at 4 °C for 10 min. The upper metabolite aqueous layer and bottom lipid organic layer were collected separately, transferred to clean glass vials, dried in a CentriVap concentrator (Labconco, Kansas City, MO) and stored at −70 °C until analysis. The protein interphase was washed by adding 1 mL of −20 °C methanol, vortexing for 1 min and centrifuging at 12,000 rpm at 4 °C for 10 min. The supernatant was discarded, and the protein pellet was dried in a vacuum centrifuge (Savant, Holbrook, NY, USA) for 5 min then stored at −70 °C until being prepared for proteomic analysis. 

#### 2.2.2. Solid-Phase Extraction of Lipids

Cell pellets were suspended in 400 μL of water and transferred to C:M (2:1, v:v) and chloroform:methanol:water C:M:W (1:2:0.8, v:v:v) prewashed polytetrafluoroethylene (PTFE)-lined 13 × 100 mm Pyrex glass tubes containing 200 μL of silica/zirconia beads. One mL of methanol and 0.5 mL of chloroform were added to each tube and lipids were extracted by vortexing for 2 min then centrifugation for 10 min at 1800× *g* at room temperature. The supernatants were collected into fresh PTFE-lined 13 × 100 mm Pyrex glass tubes. The samples were extracted with 1 mL of C:M (2:1, v:v) then 1 mL of C:M:W (1:2:0,8, v:v:v) and the supernatants were pooled together then dried in a vacuum centrifuge. Pasteur pipettes (5 ¾”, Fisher Scientific) were packed with 100 mg of silica 60 beads (Sigma, grade 7754, high purity, 70–230 mesh, 60 Å) and glass wool (Sigma) to create a sieve. The column was washed sequentially with 4 mL each of methanol, acetone and chloroform. The extracted lipids were dissolved in 1 mL of chloroform, loaded onto the column and the flow-through was collected. Sterols were eluted using 4 mL of chloroform and combined with the flow-through. Fatty acids were eluted using 4 mL of acetone and 4 mL of methanol was used to elute phospholipids. All fractions were dried in a vacuum centrifuge.

### 2.3. Proteomic Analysis

#### 2.3.1. Protein Digestion

The extracted protein fraction was suspended in 100 μL of 8 M urea in 50 mM NH_4_HCO_3_ and protein concentration was measured using the BCA protein assay (Thermo Scientific, Rockford, IL). The samples were reduced by adding 500 mM dithiothreitol (DTT) to a final concentration of 10 mM with incubation at 37 °C for 1 h with 800 rpm shaking. Iodoacetamide (400 mM) was added to a final concentration of 40 mM and incubated in the dark at 37 °C with 800 rpm shaking. Samples were diluted 8 folds with 50 mM NH_4_HCO_3_ pH 8.0. CaCl_2_ (1 M) was added to a final concentration of 1 mM and samples were digested overnight at 37 °C with 800 rpm shaking using trypsin (Thermo Fisher Scientific, Carlsbad, CA, USA) at a proportion of 1 μg trypsin/50 μg protein. After trypsin incubation, digested samples were desalted using 1 mL Discovery C18 SPE columns (Supelco, Bellefonte, PA, USA) and dried to 100 μL using a vacuum centrifuge. Peptide concentration was determined by a BCA protein assay. 

#### 2.3.2. Liquid Chromatography-Tandem Mass Spectrometry (LC-MS/MS) Analysis

Digested peptides were analyzed by liquid chromatography-tandem mass spectrometry in a Waters NanoAcquity UPLC system with a custom-packed C18 column (70 cm × 75 μm internal diameter, Phenomenex Jupiter, 3 μm particle size, 300 Å pore size) connected to a Q-Exactive mass spectrometer (Thermo Fisher Scientific, Waltham, MA, USA). Peptides were eluted with an acetonitrile gradient and analyzed online by nanoelectrospray ionization as described previously [12]. Mass spectra were collected in a range of 400–2000 m/z and 35,000 resolution at 400 m/z. The top 12 most intense parent ions were fragmented using a 2.0 m/z isolation window, 30% normalized collision energy, and 17,500 resolution at 400 m/z, before being dynamically excluded for 30 s.

#### 2.3.3. Data Analysis

Collected LC-MS/MS data were searched against the *H. capsulatum* G186AR sequences from Uniprot (downloaded 15 August 2016) using MaxQuant (Martinsried, Germany) [14]. The searches were performed considering a parent ion mass tolerance of 20 and 4.5 ppm prior and after mass recalibration, respectively. For searching, tryptic digestion in both termini was required and 2 missed cleavages allowed. Cysteine carbamidomethylation was searched as a static modification, whereas methionine oxidation and protein N-terminal acetylation were used as variable modifications. Peptide-spectral matches and proteins were both filtered with a false-discovery rate <1%. Label-free quantification method was used for quantitative analysis. This method only considered the most consistently detected peptides per protein for quantification and normalized the samples by total run intensity [15]. Significant differences between samples were determined by *t*-test considering two tails and equal variance.

Sequences were annotated according to the Kyoto Encyclopedia of Genes and Genomes (KEGG) annotation using the BlastKoala tool [16], considering only pathways present in *Saccharomyces cerevisiae*. Pathway enrichment was done by Fisher’s exact test and was filtered based on 2-fold enrichment, a minimum of 2 proteins per pathway and *p*-value ≤ 0.05. Enriched pathways were plotted into a network with the enrichment map tool using shared proteins as the connective between pathways [17]. 

### 2.4. Lipidomic Analysis

Extracted lipids were analyzed by LC-MS/MS using reverse-phase chromatography coupled to an orbitrap mass spectrometer (Orbitrap Velos with ETD, Thermo Fisher Scientific) as previously described [18]. Lipid species were identified using LIQUID and manually inspected based on the fragmentation profile and isotopic distribution for validation [19]. The features of the identified lipid species were extracted and aligned using MZmine [20].

### 2.5. Metabolomic Analysis

#### 2.5.1. Chemical Derivatization

The extracted polar metabolites and sterols were derivatized as described previously [21]. Briefly, 20 μL of 30 mg/mL methoxyamine in pyridine (Sigma-Aldrich, Saint Louis, MO, USA) was added to each sample, followed by vortexing for 10 s then incubating the samples at 37 °C for 90 min with shaking (1000 rpm). After incubation, 80 μL of N-methyl-N-(trimethylsilyl)trifluoroacetamide (MSTFA) (Sigma-Aldrich) with 1% trimethyl-chlorosilane (TMCS) (Sigma-Aldrich) was added to each sample, and the samples were vortexed for 10 s before being incubated with shaking (1000 rpm) at 37 °C for 30 min. Extracted fatty acids were methylated using 1.25 M methanolic-HCl for 2 h at 100 °C and extracted with hexane/water as previously described [22]. Derivatized samples were then transferred to autosampler vials and analyzed by GC-MS in random order. 

#### 2.5.2. GC-MS Analysis

Samples were analyzed in an Agilent GC 7890A using an HP-5MS column (30 m × 0.25 mm × 0.25 μm; Agilent Technologies, Santa Clara, CA, USA) coupled with a single quadrupole MSD 5975C (Agilent Technologies). For each analysis 1 μL of sample was injected (splitless) into the injection port with constant temperature at 250 °C. The GC analysis started with an initial oven temperature fixed at 60 °C for 1 min after injection, and then the temperature was increased to 325 °C at a rate of 10 °C/minute, and finished with a 5-min hold at 325 °C. For the background referencing and retention time calibration, blanks and fatty acid methyl-ester (FAME) samples (C8-28) were also included in the analyses, respectively. For the sterol analysis, β-sitosterol was included as internal standard and ergosterol and lanosterol as external standards.

#### 2.5.3. GC-MS Data-Processing

The raw data files were processed with Metabolite Detector as described previously [23]. In short, retention indices (RI) were calibrated based on FAME internal standards, deconvoluted and chromatographically aligned. Metabolites were identified by matching spectra and retention indices against an in-house augmented version of the FiehnLib [24] library containing validated spectral and retention indices information of more than 900 metabolites [24]. Unidentified metabolites were also searched against the NIST14 GC-MS library by comparing spectra alone. All the metabolite identifications were manually validated. The curate data were submitted to multivariate data analysis (MVDA) using MetaboAnalyst [25] by median normalization and log transforming the data to perform principal component, hierarchical cluster and heatmap analysis, which subsequently was used to identify natural clustering within the data. 

### 2.6. Susceptibility to Amphotericin B

*H. capsulatum* was cultivated for 24 h in Ham’s F12 in the presence or absence of 6B7, 7B6 and 18B7 mAb (10 µg/mL). The yeasts were then treated with increasing concentrations of Amphotericin B (AmB) for an additional 24 h. Resazurin was then added to the cells at a final concentration of 20 µM, and after 2 h at 37 °C fluorescence was measured as a viability measurement (ex: 530/em: 590). Alternatively, after the first 24 h of mAb treatment, the yeast cells were plated in the presence or absence of different concentrations of AmB and turbidity was monitored (600 nm) for up to 96 h. Selected timepoints were plated onto BHI-agar to evaluate the numbers of live cells with and without exposure to the antifungal drug.

### 2.7. Enzyme-Linked Iimmunosorbent Assay (ELISA) of Histoplasma capsulatum Cells

*H. capsulatum* yeast cells, suspended in PBS, were added to the wells of half-area high binding 96-well plates at 5 × 10^5^ cells/well at a volume of 50 μL/well and incubated overnight at 4 °C. After removing unbound yeast by washing with PBS, the plates were blocked with 2% of bovine serum albumin. Each mAb was diluted in blocking solution, serially diluted and incubated in the plates for 2 h. After washing the plate with PBS, a detection Ab (anti-mouse IgG conjugated to alkaline phosphatase) was added to the plates and incubated for one hour. After PBS washing the substrate (PNPP) was added to the plates and measured at 405 nm in a plate reader.

## 3. Results

### 3.1. Proteomics Analysis of H. capsulatum Treated with Monoclonal Antibodies

To study the effects of mAb on *H. capsulatum*, yeast cells were treated with a protective (6B7) and a non-protective (7B6) antibody, both targeting the same epitope sequence in HSP60. Control samples of *H. capsulatum* were prepared in parallel, one treated with PBS and the other treated with the same hybridoma media used to produce the mAb. GXM-binding mAb 18B7 was also used as a possible control. Treated cells were submitted to a simultaneous metabolite, protein and lipid extraction (MPLEx) consisting of the extracted proteins digested with trypsin and the resulting peptides analyzed by liquid chromatography-tandem mass spectrometry (LC-MS/MS). The analysis led to the identification of a total of 3642 proteins, of which 158 proteins were regulated by antibodies (ANOVA *p*-value ≤ 0.05 and fold change ≥ 1.5) (Figure 1A, Appendix A). Variations of different intensity of regulation were observed between the three antibody treatments; however, the trend of regulation was strikingly similar between the treatments (Figure 1A). We initially included mAb 18B7 as a negative control, but due to its similar response to the 6B7 and 7B6 mAb, we performed an ELISA experiment to test whether this antibody could also bind to *H. capsulatum*. Despite interacting to a lesser extent, 18B7 still significantly bound moieties on the surface of *H. capsulatum* yeast cells (Appendix A). We performed a function-enrichment analysis based on the KEGG annotations to better understand the processes regulated by the mAb, which showed that pathways such as biosynthesis of unsaturated fatty acids, biosynthesis of amino acids, protein export, steroid biosynthesis and fatty acid metabolism and elongation were enriched in proteins changed by the treatment (Figure 1B, Appendix A). 

### 3.2. Regulation of H. capsulatum Fatty Acid Desaturation Biosynthesis by Monoclonal Antibodies

Biosynthesis of unsaturated fatty acids was one of the most over-represented pathways (9.7X) enriched in proteins regulated by mAb compared to the whole genome background (Appendix A). Therefore, we investigated this pathway in more detail. The proteomic analysis revealed that mAb treatment reduced the levels of delta-9 fatty acid desaturase by approximately 40% but increased the levels of the oleate delta-12 desaturase by 25% (Figure 2A,B). These results suggest an increase in the levels of polyunsaturated fatty acids (Figure 2C). In order to test this hypothesis, we extracted free fatty acids using organic solvents and solid phase extraction and analyzed them by gas chromatography-mass spectrometry. As expected, the mAb treatment significantly reduced the levels of saturated fatty acid stearic acid (C18:0), while increasing the levels of the polyunsaturated fatty acid linoleic acid (C18:2) (Figure 2D). The levels of oleic acid (C18:1) had no significant differences (Figure 2D). Overall, these results showed that mAb treatment regulates the biosynthesis of unsaturated fatty acids, leading to an increase in the levels of polyunsaturated fatty acids.

### 3.3. The Impact of the Monoclonal Antibody Treatment on the Composition of Different Lipid Classes

Having shown that mAb treatment regulates the biosynthesis of unsaturated fatty acids, we investigated the impact of this regulation in the composition of different lipid classes. We performed lipidomics analysis on the MPLEx extracts, which showed a coordinated reduction of triacylglycerols (TGs) with one to two double bonds and an increase in levels of species with ≥ four double bonds (Figure 3A). In terms of phospholipids, phosphatidylcholine (PC), phosphatidylethanolamine (PE) and phosphatidylinositol (PI) all had a similar decrease in species with fewer double bonds and a coordinated increase in the species with more double bonds (Figure 3A). To validate these findings, we treated a new set of cells with mAb and submitted these samples to a different extraction protocol. Lipids were extracted with 2:1 (v:v) chloroform/methanol solution followed by a second extraction with 1:2:0.8 (v:v:v) chloroform/methanol/water solution. The extracted material was combined, dried, dissolved in pure chloroform and submitted to a solid-phase extraction in a silica 60 column. Sterols, fatty acids and phospholipids were eluted sequentially with chloroform, acetone and methanol, respectively. The analysis of the phospholipid fraction by LC-MS/MS showed a similar profile of a decrease in the more saturated species of PC, PE and PI and an increase in the levels of polyunsaturated species (Figure 3B).

Changes in other classes of lipids were also consistently observed in both lipidomics and in the analysis of the phospholipid fraction (phospholipidomics). All identified species of cardiolipin (CL) were upregulated, whereas all the inositolphosphoceramides (PI_Cer) and phosphatidylserines (PSs) were downregulated (Figure 4A,B), suggesting a regulation of the head group level for these classes of lipids. Interestingly, most of the lysophosphatidylethanolamine (LPE) and lysophosphatidylcholine (LPC) species were upregulated (Figure 4A,B), suggesting a possible increase in phospholipase activity. Like these findings, 18 out of 19 identified diacylglycerol (DG) species were increased (Figure 4A,B), further supporting the hypothesis that lipase(s) might be regulated by the mAb treatment and converting TG to DG.

Taken together, these data show that an increase in the biosynthesis of unsaturated fatty acids has an impact on the composition of TG, PC, PE and PI. The lipidomics data also show a regulation of CL, PI_Cer and PS, which seems to be driven by the regulation of head groups. Finally, the data also suggest that the higher levels of lysophospholipid species and DGs may be due to an increase in phospholipase activity.

### 3.4. Regulation of the H. capsulatum Sterol Metabolism by Monoclonal Antibodies

Another pathway of lipid metabolism highly enriched (8X) in proteins regulated by the mAb treatment was steroid metabolism (Appendix A). In this pathway, the HGM2, ERG27 and ERG5 proteins were significantly upregulated, while the ERG25 protein was downregulated by approximately 60% (Figure 5A). The remaining 15 proteins of this pathway had smaller changes or were unaltered by the treatment (Appendix A). To further investigate the impact of this regulation on the levels of sterols, we analyzed the sterol fraction of silica 60 solid-phase extraction by GC-MS. A total of 12 species of sterols were detected, of which 6 were identified, whereas another 6 species had spectral profiles which were similar between species but could not be confidently assigned (Figure 5B). The levels of ergosta-7, 22-dien-3-ol, campesterol and one unidentified species were downregulated. The levels of ergosterol, cholesterol, ergosta-7en-3-ol and 4 unidentified species were upregulated (Figure 5B). As a control, the levels of the β-sitosterol internal standard had only minor fluctuations (Figure 5B,E). Strikingly, the uptake of cholesterol only occurred in the presence of mAb (Figure 5C), despite the fact this lipid is present in the media of the control samples. The levels of the main sterol lipid in *H. capsulatum*, ergosterol, were increased by 2 folds in cells treated with mAb. Overall, these data show that antibody treatment induces changes in the sterol metabolism, increasing the levels of cholesterol and ergosterol. 

### 3.5. H. capsulatum Sensitivity to Amphotericin B in Cells Treated with Monoclonal Antibodies

The mAb treatment induced changes in different classes of lipids. Therefore, we hypothesized that such changes could affect the susceptibility of *H. capsulatum* to membrane-targeting drugs such as polyenes and azoles. To test this hypothesis, *H. capsulatum* yeasts were treated with mAb for 24 h, challenged with different concentrations of the polyene amphotericin B and checked for viability. The most evident changes were observed in higher concentrations of amphotericin B (Figure 6A). At a concentration of 1.25 μg/mL of amphotericin B, mAb 6B7 decreased the viability of *H. capsulatum* by 20%, whereas mAb 7B6 led to an approximate 60% decrease of the cell viability, compared to treated yeast that did not receive any mAb (Figure 6B). The mAb 18B7 against the *C. neoformans* capsule had no effect on the cellular viability in the presence of amphotericin B. Notably, the decrease in cell viability induced by mAbs is proportional to the binding affinity of each mAb to *H. capsulatum* cells (Appendix A) and to recombinant HSP60 [7]. The mAb 7B6 had the highest affinity and cell viability reduction, followed by 6B7 and 18B7 (Figure 6B and Appendix A). In addition, the mAb treatment by itself had no effect on fungal growth (Appendix A). These results showed that the membrane changes induced by the mAb treatment have an impact in the sensitivity of *H. capsulatum* to amphotericin B.

## 4. Discussion

In this paper, we used different mass spectrometry techniques to dissect the effects of mAbs against the surface protein HSP60 on *H. capsulatum* cells. Due to the relative lack of mAbs against *H. capsulatum* yeast cell surface molecules, as well as the lack of information about signal transduction in this yeast, we chose two mAbs that bind to the same or overlapping epitopes in surface-presented HSP60, which additionally had proven abilities to change the biological processes in this fungus [9]. Our proteomics data showed a variety of metabolic and signaling pathways that were regulated by mAb treatment. McClelland et al. showed that mAbs against capsule polysaccharides altered the expression of metabolism-associated proteins in *C. neoformans*, including four enzymes of the fatty acid biosynthesis pathways [8], and a similar pattern was observed for *H. capsulatum*. These pieces of evidence suggest a partially common response against the antibody treatment between the two yeast species, which represent two distinct phyla (Ascomycetes and Basidiomycetes). It is possible that common cellular responses, such as stress response is involved in the underlying mechanism.

Our data showed that mAbs induce changes in the biosynthesis of unsaturated fatty acids. Despite a modest 25% increase in the levels of delta-12 desaturase, a much higher impact was observed in the levels of fatty acids and the overall lipid composition. It is possible that the activity of this enzyme is regulated by post-translational events. For instance, in soybeans, delta-12 desaturase activity is inhibited by phosphorylation [26]. Fatty acid desaturases are also well-known to be regulated by temperature. In *S. cerevisiae* transcription factors Spt23 and Mga2 induce the expression of the delta-9 desaturase (Ole1), which is upregulated in low temperatures [27,28,29,30]. Conversely, in high temperatures fatty acid desaturases are targeted for degradation by the ubiquitin-proteasome system, as shown for both plants and yeast [26,31]. In agreement with these observations, the disruption of the *H. capsulatum* Ole1 gene results in an increased sensitivity to temperature changes [32,33]. Increase in unsaturated fatty acids and consequently the membrane fluidity seems to be a general mechanism of stress tolerance in yeasts by enhancing the membrane stability, ion efflux and tolerance to oxidative stress [34,35,36,37]. 

The lipidomics and phospholipidomics data showed an mAbs-dependent regulation in CL, PS and PI_Cer. The regulation seems to be targeted at the head groups since the changes were uniform within the same lipid class. Casanovas et al. described a coordinate remodeling mitochondrial lipid CL in *Saccharomyces cerevisiae* in transition from exponential to stationary phase, having a possible role in adjusting the mitochondrial function in these two distinct conditions [38]. Indeed, CL has been shown to be essential for growth with a non-fermentable carbon source, on which mitochondrial respiration is a requirement for energy production [39]. The increase in CL levels in the mAb-treated *H. capsulatum* cells suggest a regulation in mitochondrial function, possibly to boost the biosynthesis of unsaturated fatty acids and sterols. Our data also showed increased levels of LPC, LPE and DG, which could be due to higher activity of (phospho)lipases. This phenotype might be related to the lipid profile remodeling. Lysophospholipid species and DGs are intermediate steps of phospholipid and TG fatty acyl remodeling, respectively. Furthermore, PC, PE and TG are the classes of lipids most affected by the increase of unsaturated fatty acid biosynthesis and they are final products of LPC, LPE and DG re-acylation, respectively. This suggests a coordinated effect of synthesis of polyunsaturated fatty acids followed by the incorporation of more complex lipids in remodeling the plasma membrane. 

We also detected changes in the levels of sterols. We observed that media-sourced cholesterol was incorporated by the mAb-treated cells, but not by the control cells. Cholesterol uptake has not been reported in *H. capsulatum* yet. However, in *Candida glabrata*, uptake of cholesterol confers resistance to the azole class of antifungals [40,41]. Therefore, this could be a mechanism of membrane stress leading to antifungal resistance in fungi. We also found that the level of ergosterol were increased by 2 folds in *H. capsulatum* cells treated with mAbs. The regulation of ergosterol biosynthesis is poorly understood in *H. capsulatum*. However, it has been shown that this pathway is required for *H. capsulatum* adaptation to hypoxia and virulence [42]. Most of the changes in the lipid composition of *H. capsulatum* were not limited to the incubation with anti-HSP60 mAbs, as anti-GXM (18B7) also induced important modifications in the lipid composition of the yeast cells, which led to our confirming that this mAb also effectively bound to the surface of *H. capsulatum*. One possibility is that *H. capsulatum* has GXM-mimicking epitopes, possibly found as one of its surface polysaccharides [43]. It is also tempting to speculate that these cells have some mechanism of antibody recognition, which would induce changes in the lipid composition of these yeast cells in a target-independent fashion. However, this question will require further, in-depth investigation by screening a larger panel of antibodies and identification of receptor candidates. The effect of mAb binding on the susceptibility to amphotericin B was mAb-target dependent. Only anti-HSP60 mAbs affected the susceptibility of the fungus against amphotericin B. Importantly, we showed that the increased ergosterol level led to a higher susceptibility of *H. capsulatum* cells to amphotericin B, similar to what was described in *C. neoformans* [8]. This seems to be correlated to the affinity of the mAb binding to the pathogen as the mAb with the highest affinity also had the most cooperative action with amphotericin B. These observations suggest that antifungal drugs and the host immune response can cooperate in the fighting against these pathogens. 

## 5. Conclusions

In conclusion, here we used a multi-omics approach to better understand the molecular responses of *H. capsulatum* against mAbs. These findings can be extended to probable responses inside the fungus triggered by antibodies during histoplasmosis. Our data showed that antibodies induce global changes in the composition of membranes of *H. capsulatum*, increasing fungal susceptibility to amphotericin B. 

## Figures and Tables

**Figure 1 vaccines-08-00269-f001:**
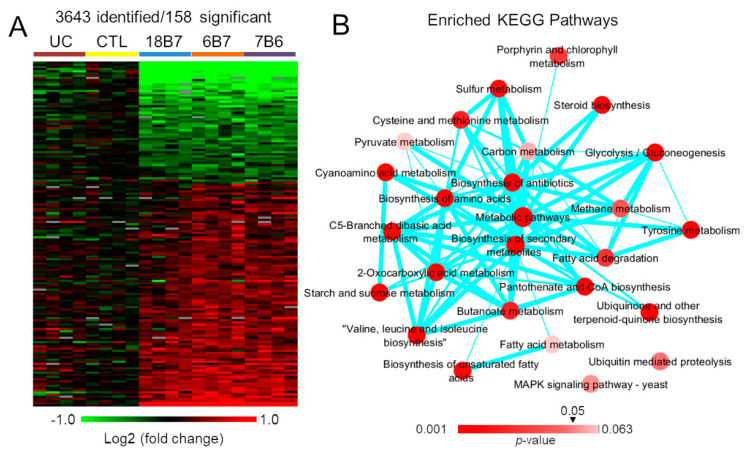
Differentially abundant proteins in *Histoplasma capsulatum* yeasts treated with monoclonal antibodies 18B7, 6B7 or 7B6 compared to medium control (CTL) and untreated control (UC) yeast cells. (**A**) Heatmap of the 158 differentially abundant proteins (ANOVA *p*-value ≤ 0.05 and fold change ≥ 1.5). (**B**) Function-enrichment analysis of the differentially abundant proteins based on the Kyoto Encyclopedia of Genes and Genomes (KEGG) annotation. The connectivity of the network was based on the shared proteins between different pathways. The intensity node color is scaled based on the enrichment *p*-values and the thickness of the edges (lines) represents the degree of shared proteins between pathways.

**Figure 2 vaccines-08-00269-f002:**
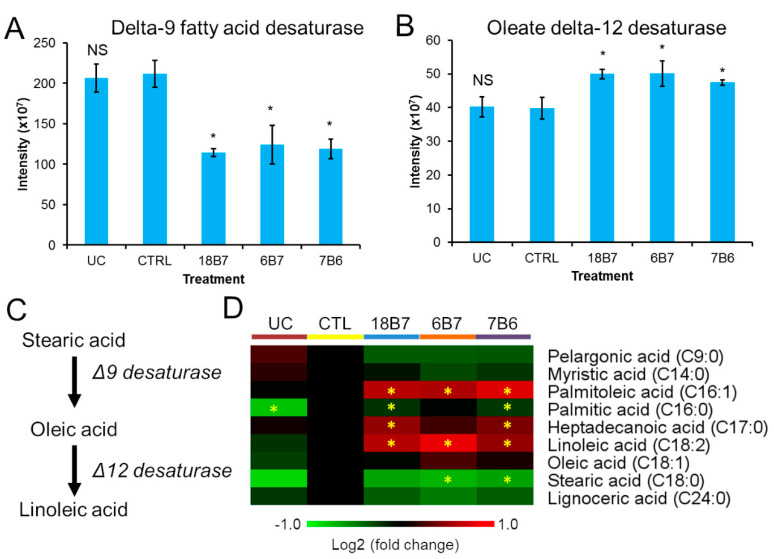
Effect of monoclonal antibodies on *H. capsulatum* unsaturated fatty acid biosynthesis. (**A,B**) Abundance profiles of delta-9 fatty acid desaturase (**A**) and oleate delta-12 desaturase in *H. capsulatum* cells treated with monoclonal antibodies. (**C**) Enzymes and metabolites of the unsaturated fatty acid biosynthesis pathway. (**D**) Abundance profiles of free fatty acids (not bound to complex lipids) in *H. capsulatum* yeast cells treated with monoclonal antibodies. Abbreviations: CTL, control; UC, untreated control without hybridoma medium; *t*-test: NS, not significant. * *p* ≤ 0.05 compared to the control sample.

**Figure 3 vaccines-08-00269-f003:**
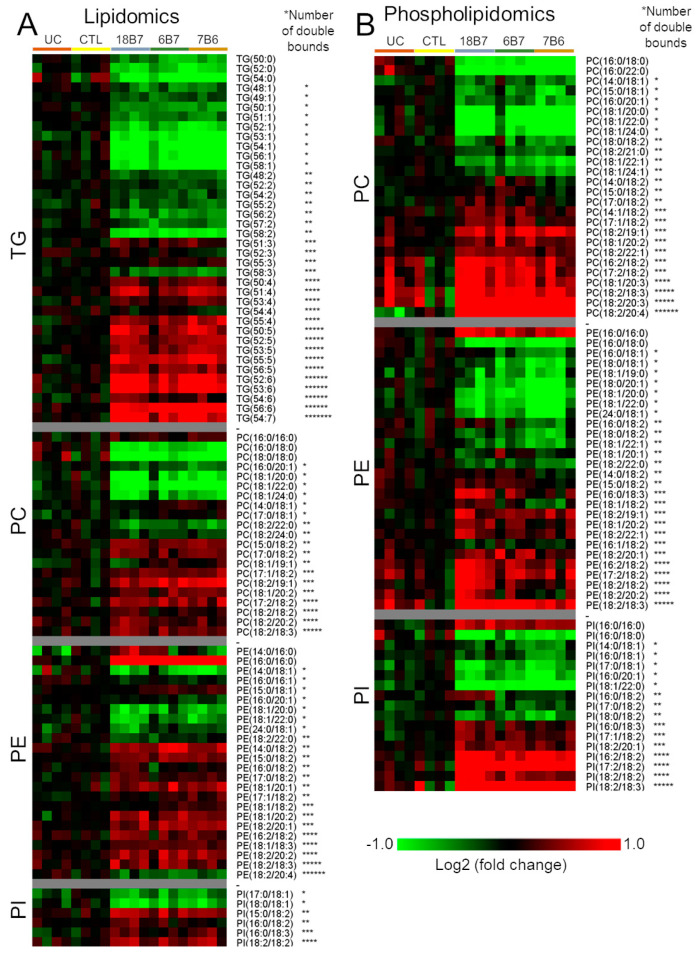
Impact of increased polyunsaturated fatty acids on the lipid composition. (**A**) Heatmap of significantly regulated triacylglycerols (TG), phosphatidylcholines (PC), phosphatidylethanolamines (PE) and phosphatidylinositols (PI) detected in the lipidomics analysis of *H. capsulatum* cells treated with monoclonal antibodies 18B7, 6B7 or 7B6. (**B**) Heatmap of significantly regulated PC, PE and PI detected in the phospholipidomics analysis of *H. capsulatum* cells treated with monoclonal antibodies 18B7, 6B7 or 7B6. Abbreviations: CTL, control; UC, untreated control without hybridoma medium. *Each asterisk represents one double bond in the fatty acyl chains.

**Figure 4 vaccines-08-00269-f004:**
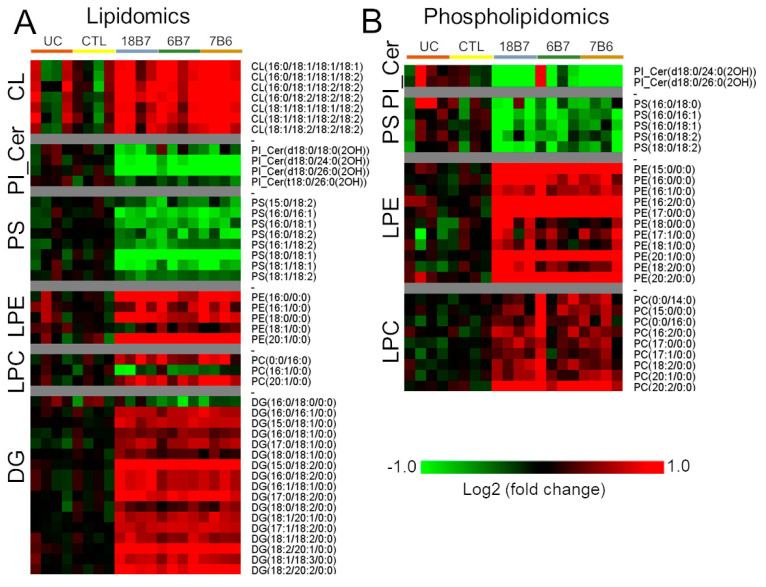
Regulation of different lipid classes in *H. capsulatum* cells by monoclonal antibody treatment. (**A**) Heatmap of significantly regulated cardiolipins (CL), inositolphosphoceramides (PI_Cer), phosphatidylserines (PS) and lysophosphatidylethanolamines (LPE), lysophosphatidylcholines (LPC) and diacylglycerols (DG) detected in the lipidomics analysis of H. capsulatum cells treated with monoclonal antibodies 18B7, 6B7 and 7B6. (**B**) Heatmap of significantly regulated PI_Cer, PS, LPE and LPC species detected in the phospholipidomics analysis of *H. capsulatum* cells treated with monoclonal antibodies 18B7, 6B7 or 7B6. Abbreviations: CTL, control; UC, untreated control without hybridoma medium.

**Figure 5 vaccines-08-00269-f005:**
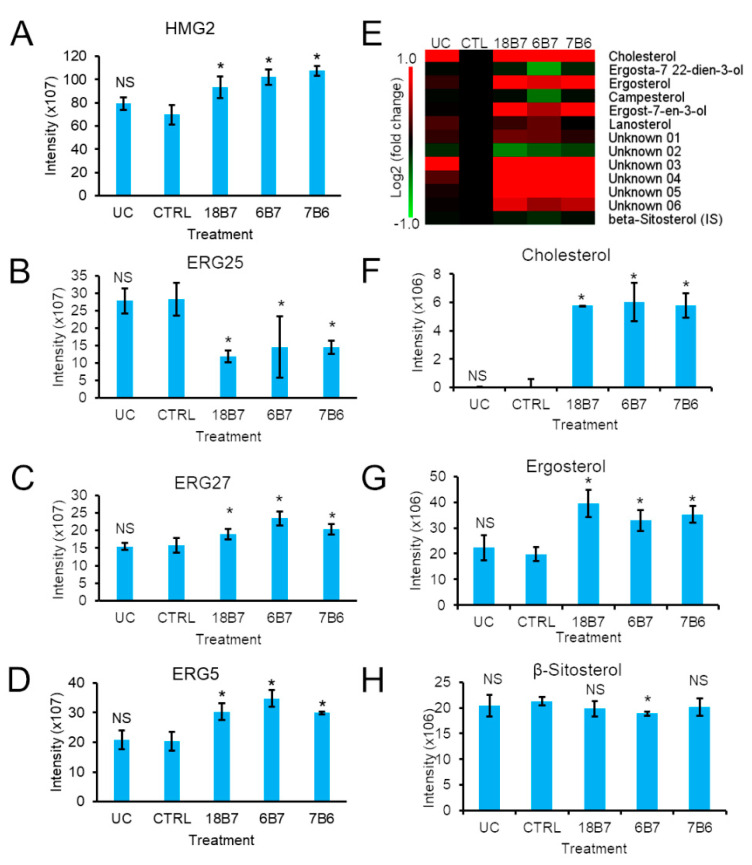
Effect of monoclonal antibodies on the *H. capsulatum* sterol metabolism. (**A**–**D**) Abundance profiles of ergosterol biosynthetic enzymes: (**A**) 3-hydroxy-3-methylglutaryl coenzyme A reductase HMG2, (**B**) C-4 methylsterol oxidase ERG25, (**C**) 3-ketosteroid reductase ERG27 and (**D**) cytochrome P450 sterol C-22 desaturase ERG5. (**E**) Heatmap of the sterol species. (**F**,**G**) Abundance profiles of sterols *H. capsulatum* cells treated with monoclonal antibodies: (**F**) cholesterol, (**G**) ergosterol and (**H**) β-sitosterol (internal standard). Abbreviations: CTRL, control; UC, untreated control without hybridoma medium. *t*-test: NS, not significant; * *p* ≤ 0.05 compared to the control sample.

**Figure 6 vaccines-08-00269-f006:**
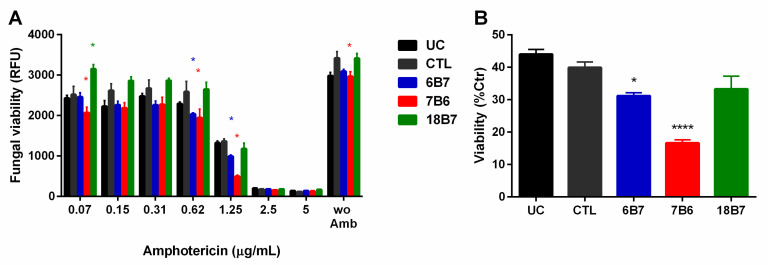
Viability of *H. capsulatum* cells pretreated with monoclonal antibodies in media containing amphotericin B. Viability was measured by fluorescence of reduced resazurin. (**A**) *H. capsulatum* yeast cells were treated or not with mAb and then incubated with different concentrations of amphotericin B. (**B**) The fungal viability for amphotericin B concentration of 1.25 µg/mL is depicted. Abbreviations: CTL, control; UC, untreated control without hybridoma medium. *t*-test: * *p* ≤ 0.05 compared to the control sample.

## Data Availability

Proteomics data were deposited into Pride repository (www.ebi.ac.uk/pride) under accession number PXD017864.

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
