# Peer review of "Remodeling of the Histoplasma Capsulatum Membrane Induced by Monoclonal Antibodies"

_vaccines, 2020, doi:10.3390/vaccines8020269_

Round 1

Reviewer 1 Report

The authors have used the multi-omics methods to map the changes in the metabolism, especially, lipid metabolism of H. capsulatum in response to antibodies against the surface protein, HSP60. The authors have presented adequate data that broadly supports their claims and I recommend that the manuscript be accepted for publication.

vaccines-78059: Remodeling of the Histoplasma capsulatum membrane induced by monoclonal antibodies
Meagan C. Burnet et al.
The authors have used the multi-omics methods to map the changes in the metabolism, especially, lipid metabolism of H. capsulatum in response to antibodies against the surface protein involved in infection, HSP60. The authors show that by changes in the lipid metabolism in presence of mAbs increased the susceptibility of these pathogens to ergosterol-targeting drug amphotericin B and these mAbs could serve as a viable therapeutic option in treatment of histoplasmosis along with amphotericin B. Overall, the results presented broadly support the claims made by the authors. I have a few following minor suggestions/comments –
1. In figure 1 and corresponding text (line 224, page 6), authors mention they used 18B7 antibody as one of the controls, however, the figure 1 legend says 1B87. The authors should correct this. The same error was observed in figure 3, and 4 legends.
2. Can authors comment/elaborate where does 18B7 binds to H. capsulatum?
3. The authors should comment/discuss the increase/decrease in levels of various lipid components studied here with respect to the two mAbs (6B7 and 7B6). Also, they should elaborate what does protective and non-protective mAbs mean?
Once these minor points are addressed and careful proofreading, I recommend that the paper be accepted for publication.

Author Response

Reviewer 1

The authors have used the multi-omics methods to map the changes in the metabolism, especially, lipid metabolism of H. capsulatum in response to antibodies against the surface protein, HSP60. The authors have presented adequate data that broadly supports their claims and I recommend that the manuscript be accepted for publication.

Thank you very much for your valuable comments.

vaccines-78059: Remodeling of the Histoplasma capsulatum membrane induced by monoclonal antibodies

Meagan C. Burnet et al.

The authors have used the multi-omics methods to map the changes in the metabolism, especially, lipid metabolism of H. capsulatum in response to antibodies against the surface protein involved in infection, HSP60. The authors show that by changes in the lipid metabolism in presence of mAbs increased the susceptibility of these pathogens to ergosterol-targeting drug amphotericin B and these mAbs could serve as a viable therapeutic option in treatment of histoplasmosis along with amphotericin B. Overall, the results presented broadly support the claims made by the authors. I have a few following minor suggestions/comments –

  1. In figure 1 and corresponding text (line 224, page 6), authors mention they used 18B7 antibody as one of the controls, however, the figure 1 legend says 1B87. The authors should correct this. The same error was observed in figure 3, and 4 legends.

R: We apologize for this typo. They have been corrected. We also performed another round of proof-reading with all the authors to minimize any typos and grammatical mistakes.

  1. Can authors comment/elaborate where does 18B7 bind to H. capsulatum?

R: Thanks for raising this point. Although we showed that 18B7 is able to bind H. capsulatum’s surface, the nature of the target is unknown. We added information to the text regarding this matter (lines 420 – 430).

Most of the changes in the lipid composition of H. capsulatum were not limited to the incubation with anti-HSP60 mAbs, as anti-GXM (18B7) also induced important modifications in the lipid composition of the yeast cells, which led to our confirming that this mAb also effectively bound to the surface of H. capsulatum. One possibility is that H. capsulatum has GXM-mimicking epitopes, possibly found as one of its surface polysaccharides [44]. It is also tempting to speculate that these cells have some mechanism of antibody recognition, which would induce changes in the lipid composition of these yeast cells in a target-independent fashion. However, this question will require further, in-depth investigation by screening a larger panel of antibodies and identification of receptor candidates. The effect of mAb binding on the susceptibility to amphotericin B was mAb-target dependent. Only anti-HSP60 mAbs affected the susceptibility of the fungus against amphotericin B.”

  1. The authors should comment/discuss the increase/decrease in levels of various lipid components studied here with respect to the two mAbs (6B7 and 7B6). Also, they should elaborate what does protective and non-protective mAbs mean?

R: We agree that the discussion was somehow too simplified. We added more information about the cardiolipin regulation, its possible function in mitochondrial function and downstream effect on lipid biosynthesis. We also more clearly tied the different levels of deacylated lipids with remodeling and incorporation of newly synthesized unsaturated fatty acids (lines 399-405).

“Casanovas et al. described a coordinate remodeling mitochondrial lipid CL in Saccharomyces cerevisiae in transition from exponential to stationary phase, having a possible role in adjusting the mitochondrial function in these two distinct conditions [39]. Indeed, CL has been shown to be essential for growth with non-fermentable carbon source, on which mitochondrial respiration is a requirement for energy production [40]. The increase in CL levels in the mAb-treated H. capsulatum cells suggest a regulation in mitochondrial function, possibly to boost the biosynthesis of unsaturated fatty acids and sterols.”

Regarding the protective/non-protective nature of anti-HSP60 mAbs, more information was added to the text in order to clarify the mAbs’ role in the experimental infection of mice with H. capsulatum (lines 63 – 66).

“We selected two mAbs that bind the same HSP60 epitope on the surface of H. capsulatum. Treatment of mice with these mAbs leads to different outcomes in infection with H. capsulatum. 6B7 (IgG1) confers significant protection in a lethal histoplasmosis model in mice, whereas 7B6 (IgG2b) is not protective [7].”

Reviewer 2 Report

Burnet et al presented an exhaustive study Histoplasma capsulatum response to monoclonal antibody. The authors carefully work through the  changes in the yeast membrane exploiting thee multi-omics data. The authors further study the effect of Amphotericin B treatment.  I have a few minor comments.

  1. The figures are text in the figures are not clear and sometimes eligible. Also the figures are low resolution.
  2. Figure 6 a, y-axis label is missing. The fonts and font sizes in FIG 6 does not match.
  3. The authors should discuss the limitations of the study

Author Response

Reviewer 2

Burnet et al presented an exhaustive study Histoplasma capsulatum response to monoclonal antibody. The authors carefully work through the changes in the yeast membrane exploiting thee multi-omics data. The authors further study the effect of Amphotericin B treatment.  I have a few minor comments.

Thank you very much for your valuable comments.

The texts in the figures are not clear and sometimes eligible. Also the figures are low resolution.

Figure 6 a, y-axis label is missing. The fonts and font sizes in FIG 6 does not match.

R: Thanks for pointing this. The axis and the fonts in Figure 6A were changed to be uniform in format. To clarify, the journal guidelines ask to incorporate the figures in the text in the first submission. We are now submitting the high-resolution files to address this concern.

The authors should discuss the limitations of the study

R: We explained why we choose those mAbs as a limitation of the work (lines 373 – 376).

“Due to the relative lack of mAbs against H. capsulatum yeast cell surface molecules, as well as the lack of information about signal transduction in this yeast, we chose two mAbs that bind to the same or overlapping epitopes in surface presented HSP60, which additionally had proven abilities to change biological processes in this fungus”.

Also (lines 427 – 428):

“Most of the changes in the lipid composition of H. capsulatum were not limited to the incubation with anti-HSP60 mAbs, as anti-GXM (18B7) also induced important modifications in the lipid composition of the yeast cells, which led to our confirming that this mAb also effectively bound to the surface of H. capsulatum. One possibility is that H. capsulatum has GXM-mimicking epitopes, possibly found as one of its surface polysaccharides [44]. It is also tempting to speculate that these cells have some mechanism of antibody recognition, which would induce changes in the lipid composition of these yeast cells in a target-independent fashion. However, this question will require further, in-depth investigation by screening a larger panel of antibodies and identification of receptor candidates.”

Reviewer 3 Report

The manuscript Remodeling of the Histoplasma capsulatum membrane induced by monoclonal antibodies, explores the pathogen response to mAb binding, specifically focussing on the impact to fatty acid biosynthesis pathways.

The paper is well written and on the whole, well presented. The novelty of this work however lies purely in the metabolomic analysis of fatty acid pathways, which is detracted from by the limited analysis of augmented amphotericin B toxicity. Work that has been demonstrated many times before, including by some of the authors of this paper.

Specific points:

1. The introduction is quite short given the extensive history of this approach to anti-fungal therapy. It could easily be expanded and misses recent publications of relevance. 

2. Some of the figures need revising as they are difficult to read and labels are missing/misplaced. The image quality of the figures needs improving.

3. The conclusions are over stated in relation to the impact of mAb on amphotericin B treatment. In this reviewer's opinion, this data adds nothing to the paper and should be removed. However, if it is kept in additional analysis is needed. For example, is there a difference in fungal cell viability between the antibody treatments for example, and why do the authors think that there was only an impact at a concentration of 1.25ug/ml?

4. A major limitation of this study is the absence of a proper control mAb. As stated by the authors, 18B7 was intended as a negative control, but found to bind H. capsulatum. This severely limits the conclusions drawn because it is not possible to determine whether the fatty acid changes are due to binding to HSP60 or non-specific effects. Additional antibody binding kinetics data (e.g. through SPR) to supplement the ELISA data would be useful. It would also be prudent to include a greater number of antibodies in the experiments, including antibodies to the membrane proteins and a proper negative control.

5. The work is of interest, but more clarity is needed in the discussion/conclusions regarding the relevance of the findings for treatment of fungal infections. A more in-depth discussion of the plethora of factors that could impact the downstream effects of antibody binding including antibody isoptype, binding kinetics, metabolic stress etc. is needed.

Author Response

Reviewer 3

The manuscript Remodeling of the Histoplasma capsulatum membrane induced by monoclonal antibodies, explores the pathogen response to mAb binding, specifically focusing on the impact to fatty acid biosynthesis pathways.

The paper is well written and on the whole, well presented. The novelty of this work however lies purely in the metabolomic analysis of fatty acid pathways, which is detracted from by the limited analysis of augmented amphotericin B toxicity. Work that has been demonstrated many times before, including by some of the authors of this paper.

R: We agree that the interaction between the metabolism and antifungal drugs have been demonstrated in other papers. However, that is not the focus of our paper. In this paper we focused on the fungal responses to antibody treatment, which led us to further investigate the lipid metabolism. We would like to emphasize that there are more information about the roles of antibodies in fighting infections and the interaction between the cellular metabolism and drugs, but very little is known about the fungal responses to the host antibodies.

Specific points:

  1. The introduction is quite short given the extensive history of this approach to anti-fungal therapy. It could easily be expanded and misses recent publications of relevance.

R: We agree that more can be said about anti-fungal therapy. However, we didn’t want to give an extensive review of this subject because it is a little off the focus of the paper, which is on the fungal responses to the antibodies. We added the following information about the subject to the text (lines 53 – 57).

“Recently, engineered chimeric antibodies have been shown to confer pan-fungal protection by targeting chitin, a conserved polysaccharide present across the Fungi Kingdom [5]. In Histoplasma capsulatum, antibodies against the cell-wall proteins, histone 2B-like protein and 60 kDa heat shock protein (HSP60), are protective by opsonizing and altering the intracellular fate of the fungus inside macrophages, leading to the production of Th1 type cytokines.”

  1. Some of the figures need revising as they are difficult to read and labels are missing/misplaced. The image quality of the figures needs improving.

R: Thanks for raising this concern. To clarify, the journal guidelines ask to incorporate the figures in the text in the first submission. We are now submitting the high-resolution files to address this concern.

  1. The conclusions are over stated in relation to the impact of mAb on amphotericin B treatment. In this reviewer's opinion, this data adds nothing to the paper and should be removed. However, if it is kept in additional analysis is needed. For example, is there a difference in fungal cell viability between the antibody treatments for example, and why do the authors think that there was only an impact at a concentration of 1.25ug/ml?

R: We agree with the reviewer that the effects are modest to some level. However, we performed this experiment 4 independent times to make sure that this is significant. The 1.25 ug/mL concentration was depicted to show where the effect of mAbs was critical. However, Fig. 6A shows other amphotericin B concentrations and the respective effects on H. capsulatum treated with mAbs. Some effects induced by 18B7 are similar to those induced by anti-hsp60 mAb, but 18B7 did not change the susceptibility of H. capsulatum against amphotericin B. In a lower concentration of amphotericin B (0.07 µg/mL), 18B7 conferred some resistance to the fungus. We believe that this is an important piece of information, as it shows that mAbs may induce effects in binding site-dependent and independent fashions. We are also including data on the effect of mAbs on fungal cell viability. Our data shows that up to 96 hours after the incubation with the mAbs, no significant change in fungal cell viability was observed (Figure S2).

Figure S2. Histoplasma capsulatum growth in the presence of monoclonal antibodies. Cells were grown in Ham’s F12 in the presence or absence of 6B7, 7B6 and 18B7 mAb (10 µg/mL). Growth was monitored by absorbance at 600 nm in a plate reader.

  1. A major limitation of this study is the absence of a proper control mAb. As stated by the authors, 18B7 was intended as a negative control, but found to bind H. capsulatum. This severely limits the conclusions drawn because it is not possible to determine whether the fatty acid changes are due to binding to HSP60 or non-specific effects. Additional antibody binding kinetics data (e.g. through SPR) to supplement the ELISA data would be useful. It would also be prudent to include a greater number of antibodies in the experiments, including antibodies to the membrane proteins and a proper negative control.

R: We agree with the reviewer’s point. We initially expected 18B7 to be an innocuous mAb. However, our data show that, to some extent, 18B7 also led to changes similar to that derived with anti-HSP60 mAbs. We suggested that the yeast cells could have a mechanism to sense IgG binding and signal as a cell-wall stressor leading to changes in lipid composition, but regarding susceptibility changes to amphotericin B the mAb’s specificity plays a role, as 18B7 had no effect.

We choose the two HSP60-binding mAbs based on a previous publication showing the ability of these mAbs to induce changes in the loading of exosomes released by H. capsulatum. The screening of a larger repertoire of mAbs would be a very interesting experiment. Because we do not have a functional screening assay, this experiment would difficult, laborious and costly to do. ELISA experiments would allow us to identify antibodies that either bind or not to H. capsulatum cells. Unfortunately, the antibody-amphotericin B co-treatment assay cannot address if the additional antibodies have an effect on the cells and we would need to perform proteomics or lipidomics analysis to test this, which would cost thousands of dollar per sample and take several months to over a year to be done. With the current funding and pandemic scenario (our labs are working with only about 10% of the staff), this is unfortunately not feasible to do. We also want to emphasize that this experiment would bring additional insights on whether or not H. capsulatum has a mechanism for antibody recognition, but not change the conclusions of the current paper.

To acknowledge that in the paper, we added the following sentences to the text (lines 373 – 376).

“Due to the relative lack of mAbs against H. capsulatum yeast cell surface molecules, as well as the lack of information about signal transduction in this yeast, we chose two mAbs that bind to the same or overlapping epitopes in surface presented HSP60, which additionally had proven abilities to change biological processes in this fungus”.

Also (lines 427 – 428):

“Most of the changes in the lipid composition of H. capsulatum were not limited to the incubation with anti-HSP60 mAbs, as anti-GXM (18B7) also induced important modifications in the lipid composition of the yeast cells, which led to our confirming that this mAb also effectively bound to the surface of H. capsulatum. One possibility is that H. capsulatum has GXM-mimicking epitopes, possibly found as one of its surface polysaccharides [44]. It is also tempting to speculate that these cells have some mechanism of antibody recognition, which would induce changes in the lipid composition of these yeast cells in a target-independent fashion. However, this question will require further, in-depth investigation by screening a larger panel of antibodies and identification of receptor candidates.”

  1. The work is of interest, but more clarity is needed in the discussion/conclusions regarding the relevance of the findings for treatment of fungal infections. A more in-depth discussion of the plethora of factors that could impact the downstream effects of antibody binding including antibody isoptype, binding kinetics, metabolic stress etc. is needed.

R: We agree with the reviewer’s point and added more information to the discussion section. It is not well known which immediate downstream changes are induced by these mAbs, especially regarding intracellular signaling. We previously published information about the dissociation constants of the anti-HSP60 mAbs as well as the affinity for HSP60. Besides the affinity of mAbs against the yeast cell, we referenced the data about the affinity of anti-HSP60 mAbs against HSP60 (lines 358-360). We are currently working on the change of isotype of the anti-HSP60 mAbs, that will be the subject of a future work. As we stated above, we would love to do more experiments, which are in our plans, but they are not feasible in the current scenario.

“Notably, the decrease in cell viability induced by mAbs is proportional to the binding affinity of each mAb to H. capsulatum cells (Figure S1) and to recombinant HSP60 [7].”

Round 2

Reviewer 3 Report

The modifications made by the authors address all of the previous queries raised. This is an interesting and well written manuscript that will be of interest to the field.